# Interventional Rationalization

**Linan Yue**[1,2], **Qi Liu**[1,2,*] **Li Wang**[1,2,3], **Yanqing An**[1,2], **Yichao Du**[1,2], **Zhenya Huang**[1,2]

1: Anhui Province Key Laboratory of Big Data Analysis and Application,
University of Science and Technology of China
2: State Key Laboratory of Cognitive Intelligence
3: ByteDance
{lnyue,wl063,anyq,duyichao}@mail.ustc.edu.cn;
{qiliuql,huangzhy}@ustc.edu.cn

## Abstract

Selective rationalizations improve the explainability of neural networks by selecting a subsequence of the input (i.e., rationales) to explain the prediction results. Although existing methods have achieved promising results, they still suffer from adopting the spurious correlations in data (aka., shortcuts) to compose rationales and make predictions. Inspired by the causal theory, in this paper, we develop an interventional rationalization (Inter-RAT) to discover the causal rationales. Specifically, we first analyse the causalities among the input, rationales and results with a causal graph. Then, we discover spurious correlations between the input and rationales, and between rationales and results, respectively, by identifying the confounder in the causalities. Next, based on the backdoor adjustment, we propose a causal intervention method to remove the spurious correlations between input and rationales. Further, we discuss reasons why spurious correlations between the selected rationales and results exist by analysing the limitations of the sparsity constraint in the rationalization, and employ the causal intervention method to remove these correlations. Extensive experimental results on three real-world datasets clearly validate the effectiveness of our proposed method. The source code of Inter-RAT is available at https://github.com/yuelinan/Codes-of-Inter-RAT.

## 1 Introduction

The remarkable success of deep neural networks (DNNs) in natural language processing tasks has prompted the interest in how to explain the results of DNNs. Among them, the selective rationalization task (Lei et al., 2016; Yu et al., 2019, 2021) has received increasing attention, answering the question "What feature has a significant impact on the prediction results of the model?". Specifically, the goal of rationalization is to extract a small

---

\* Corresponding Author

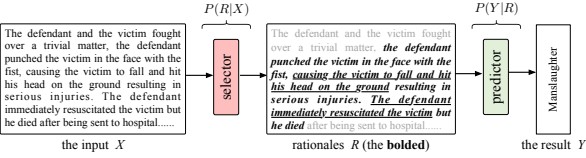

Figure 1: Conventional framework of rationalization presented in this paper. In the charge prediction, the input $X$ represents the case fact and the result $Y$ denotes the charge.

subset of the input (i.e., rationale) to support and explain the prediction results when yielding them. Existing methods often generate rationales with a conventional framework consisting of a *selector* (aka., *rationale generator*) and a *predictor* (Lei et al., 2016). As shown in Figure 1, giving the input $X$, the *selector* and the *predictor* generate rationales $R$ and prediction results $Y$ cooperatively (i.e., $P(Y|X) = P(Y|R)P(R|X)$). Among them, the *selector* ($P(R|X)$) first extracts a subsequence of input $R$. Then, the *predictor* ($P(Y|R)$) yields results based only on the selected tokens, and the selected subsequence is defined as the rationale.

Despite the appeal of the rationalization methods, the current implementation is prone to exploit spurious correlations (aka., shortcuts) between the input and labels to yield the prediction results and select the rationales (Chang et al., 2020; Wu et al., 2022). We illustrate this problem with an example of the charge prediction[1]. Considering Figure 1, although this case is corresponding to the *Manslaughter*, a DNNs model readily predicts the charge as *Intentional homicide*. Specifically, as *Intentional homicide* occurs more frequently than *Manslaughter*[2] and is often accompanied by tokens denoting violence and death, DNNs do not need to learn the real correlations between the case facts and the charge to yield the result. Instead, it is much easier to exploit spurious correlations in data

---

[1]Charge prediction: predicting the charge such as *Robbery* and *Theft* based on the case fact. Detailed definition of charge prediction is described in section 4.3.

[2]https://wenshu.court.gov.cn

to achieve high accuracy (i.e., predicting the charge as *Intentional homicide* directly when identifying the tokens about violence and death.). As a result, when facing the cases such as the example in Figure 1, the effectiveness of such DNNs tends to degrade (e.g., the underlined tokens in Figure 1 denoting the offence is negligent will be ignored in rationales extraction and the charge will be misjudged.). Therefore, these type DNNs depending on spurious correlation in data fail to reveal truly critical subsequence for predicting labels.

To solve that, (Chang et al., 2020) propose an environment-invariant method (INVRAT) to discover the causal rationales. They argue that the causal rationales should remain stable as the environment shifts, while the spurious correlation between input and labels vary. Although this method performs well in selecting rationales, since the environment in rationalization is hard to observe and obtain, we argue that this "*causal pattern*" can be further explored to improve the rationalization.

Along this research line, in this paper, we propose an interventional rationalization (Inter-RAT) method which removes the spurious correlation by the causal intervention (Glymour et al., 2016). Specifically, motivated by the causal inference theory, we first formulate the causal relationships among $X$, $R$ and $Y$ in a causal graph (Pearl et al., 2000; Glymour et al., 2016) as shown in Figure 2(a). Then, we identify the confounder $C$ in this causal graph, which opens two backdoor paths $X \leftarrow C \rightarrow R$ and $R \leftarrow C \rightarrow Y$, making $X$ and $R$, $R$ and $Y$ spuriously correlated. Next, we address the above correlations, respectively. For spurious correlations between $X$ and $R$, we assume the confounder is observed and intervene the $X$ (i.e., calculating $P(R|do(X))$ instead of $P(R|X)$) to block the backdoor path and remove the spurious correlations based on the backdoor adjustment (Glymour et al., 2016). Among them, the *do*-operation denotes the pursuit of real causality from $X$ to $R$. For spurious correlations in $R$ and $Y$, since by the definition of $R$ (rationales are the only basis for yields prediction results), we argue that there should be no spurious correlations between $R$ and $Y$. However, in practice, we discover the sparsity constraint commonly defined in rationalization (Lei et al., 2016; Cao et al., 2020; Chang et al., 2020; Yu et al., 2019), ensuring the *selector* to extract short rationales, results in the spurious correlations between $R$ and $Y$. Therefore,

we further analyse this discovery and employ the causal intervention to remove these correlations. Our experiments are conducted on three real-world datasets, and the experimental results validate the effectiveness of removing spurious correlation with causal interventions.

## 2 Framework of Rationalization

This section formally defines the problem of rationalization, and then presents the details about the conventional rationalization framework consisting of the *selector* and *predictor*, where these two components are trained cooperatively to generate rationales and yield the prediction results.

### 2.1 Problem Formulation

Considering a text classification task, only the text input $X = \{x_1, x_2, \ldots, x_n\}$, where $x_i$ represents the $i$-th token, and the discrete ground truth $Y$ are observed during training, while the rationale $R$ is unavailable. The goal of selective rationalization is first adopting the *selector* to learn a binary mask variable $M = \{m_1, m_2, \ldots, m_n\}$, where $m_j \in \{0, 1\}$, and further select a subsequence of input $R = M \odot X = \{m_1 \cdot x_1, m_2 \cdot x_2, \ldots, m_n \cdot x_n\}$, and then employing the *predictor* to re-recode the mask input $R$ to yield the results. Finally, the whole process of rationalization is defined as:

$$P(Y|X) = \underbrace{P(Y|R)}_{predictor} \underbrace{P(R|X)}_{selector}. \qquad (1)$$

### 2.2 Selector

The *selector* divides the process of generating rationales into three steps. First, the *selector* samples each binary value $m_j$ from the probability distribution $P(\widetilde{M}|X) = \{p_1, p_2, \ldots, p_n\}$, where $p_j$ represents the probability of selecting each $x_j$ as the part of the rationale. Specifically, $p_j$ is calculated as $p_j = P(\widetilde{m}_j|x_j) = \text{softmax}(W_e f_e(x_j))$, where the encoder $f_e(\cdot)$ encodes the token $x_j$ into a d-dimensional vector and $W_e \in \mathbb{R}^{2 \times d}$. Then, to ensure the sampling operation is differentiable, we adopt the Gumbel-softmax method (Jang et al., 2017) to achieve this goal:

$$m_j = \frac{\exp\left((\log(p_j) + g_j)/\tau\right)}{\sum_t \exp\left((\log(p_t) + g_t)/\tau\right)}, \qquad (2)$$

where $\tau$ is a temperature hyperparameter, $g_j = -\log(-\log(u_j))$ and $u_j$ is random sampled from the uniform distribution $U(0, 1)$. Finally, the rationale can be selected as $R = M \odot X =$

$\{m_1 \cdot x_1, m_2 \cdot x_2, \ldots, m_n \cdot x_n\}$. Therefore, we conclude that the probability of generating rationales $P(R|X)$ is calculated as: $P(R|X) = P(M \odot X|X) \equiv P(\widetilde{M}|X)$.

## 2.3 Predictor

Based on selected rationale tokens $R$, the *predictor* outputs the prediction results (i.e., calculating $P(Y|R) = P(Y|M \odot X)$), and then $R$ can be seen as an explanation of $Y$. Specifically, after obtaining $R$ from the *selector*, we adopt the neural network $f_p(\cdot)$ to re-encode the rationale into d-dimensional continuous hidden states to yield results. The objective of the *predictor* is defined as:

$$\mathcal{L}_{task} = \mathbb{E}_{\substack{X,Y \sim \mathcal{D}_{tr} \\ M \sim P(\widetilde{M}|X)}} \left[ \ell\left(Y, W_p f_p\left(M \odot X\right)\right) \right], \tag{3}$$

where $\mathcal{D}_{tr}$ denotes the training set, $\ell(\cdot)$ represents the cross-entropy loss function, $W_p \in \mathbb{R}^{N \times d}$ is the trained parameter and $N$ is the number of labels (e.g., $N = 2$ in the binary classification).

## 2.4 Sparsity and Continuity Constraints

Since an ideal rationale should be a short and coherent part of original inputs, we add the sparsity and continuity constraints (Lei et al., 2016):

$$\mathcal{L}_{re} = \lambda_1 \left| \frac{1}{n} \sum_{j=1}^{n} m_j - \alpha \right| + \lambda_2 \sum_{j=2}^{n} |m_j - m_{j-1}|, \tag{4}$$

where the first term encourages the model to select short rationales and $\alpha$ is a predefined sparsity level at the scale of [0, 1], and the second term ensures the coherence of selected tokens. Finally, the overall objective of the rationalization is defined as: $\mathcal{L} = \mathcal{L}_{task} + \mathcal{L}_{re}$.

## 3 Interventional Rationalization

In this section, we first reveal how the confounder $C$ causes spurious correlations in rationalization with a causal graph. Then, we remove these correlations by using a causal intervention method.

## 3.1 Causal Graph in Rationalization

As shown in Figure 2(a), we formulate the causalities among the text input $X$, rationale $R$, ground-truth label $Y$ and the confounder $C$ with a causal graph (Pearl et al., 2000; Glymour et al., 2016), where the link between two variables represents a causal relationship. In this paper, we only show

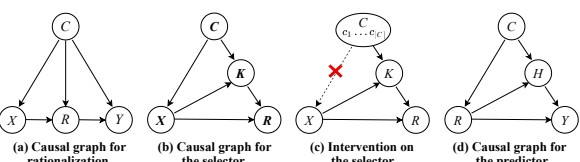

(a) Causal graph for rationalization.    (b) Causal graph for the selector.    (c) Intervention on the selector.    (d) Causal graph for the predictor.

Figure 2: (a) Causal graph for rationalization. (b) Causal graph for the selector, which offers a fine-grained causal relationship between $R$ and $C$ with a mediator $K$. (c) Intervention on the selector, where $C \to X$ is cut off and the confounder $C$ is stratified into pieces $C = \{c_1, c_2, \ldots, c_{|C|}\}$. (d) Causal graph for the predictor, which describes a fine-grained causal relationship between $Y$ and $C$ with a mediator $H$.

the endogenous variables we are interested in, following the settings in (Fan et al., 2022). In the following, we introduce the causal graph with these variables at a high-level:

$C \to X$. The confounder $C$ in rationalization can be seen as the context prior, determining which tokens can be "put" into the text input $X$. Among them, in practice, the context prior consists of unobserved prior and observed prior (e.g., we consider the entire label set as the partially observed confounder in the text classification. See section 3.2 for details.) We take the Figure 1 as an example to further illustrate $C \to X$. Specifically, both observed *Intentional homicide* and *Manslaughter* priors decide where the tokens denoting violence and death appear ; the *Manslaughter* prior determines where the tokens representing manslaughter appear ; the unobserved prior decides where other tokens that are meaningless appear.

$X \to R \leftarrow C$. Besides the *selector* extracts a subsequence of $X$ as the rationale $R$, making $X \to R$ holds, $R$ is also affected by the context prior $C$. Figure 2(b) offers a fine-grained causal relationship between $R$ and $C$ with a mediator $K$. Specifically, in $X \to K \leftarrow C$, $K$ denotes the context-specific representation which is a weighted representation of the prior knowledge associated with $X$ in $C$. In $X \to R \leftarrow K$, $R$ is affected by the context prior $C$ through $K$ indirectly. For example, in Figure 1, the underlined tokens denoting negligent are ignored in rationalization, since the context prior in $C$ misleads the model to focus on the violence and death feature in $X$ by the mediator $K$. Detailed examples about explanations for this causal graph can be found in Appendix A.1.

$C \to Y \leftarrow R$. As the *predictor* yields the result based on the rationale, $R \to Y$ holds. Meanwhile, in the ideal situation, since the rationale is defined as a subsequence of $X$ sufficient to pre-

dict the $Y$, there should be no direct causal relationship between $C$ and $Y$. However, in practice, rationales $R$ are commonly extracted with shortcut tokens (we will introduce it later in section 3.3), making $C \rightarrow Y$ exists. Figure 2(d) describes a fine-grained causal graph between $Y$ and $C$ with a mediator $H$, where $H$ is the context-specific representation of $R$ by using the context prior $C$. More detailed descriptions of this causal graph are presented in Appendix A.2.

From the graph, we find that $X$ and $R$, $R$ and $Y$ are confounded by the context prior $C$ with two backdoor paths $X \leftarrow C \rightarrow R$ (or $X \leftarrow C \rightarrow K \rightarrow R$ for elaboration) and $R \leftarrow C \rightarrow Y$ (or $R \leftarrow C \rightarrow H \rightarrow Y$). The above backdoor paths result in spurious correlations among the text input $X$, rationale $R$, and label $Y$. Based on this, we propose a causal intervention method to remove the confounding effect by cutting off the link $C \rightarrow X$ and $C \rightarrow R$, respectively.

## 3.2 Backdoor Adjustment

To pursue the real causality from $X$ to $R$ (or $R$ to $Y$), we adopt the causal intervention $P(R|do(X))$ instead of $P(R|X)$ (or $P(Y|do(R))$ instead of $P(Y|R)$) to remove the effects of confounder $C$. Next, we introduce the causal intervention method by taking $P(R|do(X))$ as an example, and $P(Y|do(R))$ is similar. Specifically, since adopting the randomized controlled trial to intervene $X$ is impossible, which requires the control over causal features, we apply the backdoor adjustment (Glymour et al., 2016) to achieve $P(R|do(X))$ by cutting off $C \rightarrow X$ (Figure 2(c)):

$$P(R|do(X)) = \sum_{i=1}^{|C|} \left[ P\left(R|X, K = g_s\left(X, c_i\right)\right) P\left(c_i\right) \right],$$
(5)

where the confounder $C$ is stratified into pieces $C = \{c_1, c_2, \ldots, c_{|C|}\}$, $P(c_i)$ denotes the prior distribution of $c_i$, which is calculated before training, and $g_s(\cdot)$ is a function achieving $X \rightarrow K \leftarrow C$. However, the confounder $C$ is commonly hard to observe. Fortunately, based on the existing researches (Wang et al., 2020; D'Amour, 2019), we can consider the entire label set as the partially observed children of the unobserved confounder. Therefore, we approximate it by designing a dictionary $D_c = \{c_1, c_2, \ldots, c_{|N|}\}$ as an $N \times d$ matrix, where $N$ represents the number of labels and $d$ is the hidden feature dimension. As described in section 2.2, we conclude

$P(R|X) \equiv P(\widetilde{M}|X)$. Therefore, we can achieve $P(R|do(X)) \equiv P(\widetilde{M}|do(X))$. Specifically, to calculate the probability of each token $x_j$ selected as the rationale, the implementation is defined as:

$$
\begin{aligned}
P(\widetilde{m}_j|do(X)) &= \sum_{i=1}^{|N|} \left[ P\left(\widetilde{m}_j|f_s(x_j, k_i)\right) P\left(c_i\right) \right] \\
&= \sum_{i=1}^{|N|} \left[ \text{softmax}(f_s(x_j, k_i)) P\left(c_i\right) \right].
\end{aligned}
$$
(6)

Among them, $f_s(\cdot)$ is the function achieving $X \rightarrow R \leftarrow K$, $k_i \in K$ is defined as the content-specific representation by using the context prior $c_i$, we express it as $k_i = g_s(x_j, c_i) = \lambda_i c_i$, where $\lambda_i \in \lambda$. Besides, $\lambda \in \mathbb{R}^N$ is the set of the normalized similarity between $x_j$ and each $c_i$ in the confounder set $C$ (i.e., $\lambda = \text{softmax}(f_e(x_j) D_c^T)$). Furthermore, since Eq (6) requires sampling of $C$ and this sampling is expensive, we try to find an alternative function that would be easy to compute to approximate it. Empirically, based on the results in (Xu et al., 2015; Wang et al., 2020; Yue et al., 2020), we can adopt the NWGM approximation (Xu et al., 2015) to move the outer sum into the softmax: $P(\widetilde{m}_j|do(X)) \approx \text{softmax}(\sum_{i=1}^{|N|} f_s(x_j, k_i) P\left(c_i\right))$.

In this paper, we adopt the linear model $f_s(x_j, k_i) = W_1 f_e(x_j) + W_2 k_i = W_1 f_e(x_j) + W_2 \lambda_i c_i$ to fuse the information of the input $X$ and the confounder $C$. Then, the final implementation of the intervention is formulated as:

$$
\begin{aligned}
P(\widetilde{m}_j|do(X)) = \text{softmax}(W_1 f_e(x_j) \\
+ W_2 \sum_{i=1}^{|N|} \lambda_i c_i P\left(c_i\right)).
\end{aligned}
$$
(7)

## 3.3 Limitations on the Predefined Sparsity $\alpha$

In this section, we discuss why $C \rightarrow Y$ in Figure 2(a) holds in detail. Since rationales are defined as the subsequence of inputs, being sufficient to yield results, $C \rightarrow Y$ should not exist. However, unfortunately, in practical implementation, the sparsity constraint (denoted by $\alpha$-constraint) in the first term of Eq (4) may result in spurious correlations between the extracted rationale and the predicted result. Specifically, the $\alpha$-constraint encourages the *selector* to extract $\alpha$ of tokens from the original text input. When the predefined number of extracted tokens is greater than the length of the practical rationale, a few tokens corresponding to

shortcuts of $Y$ may still be selected. For example, as $\alpha$ converges to 1, all tokens in the input will be extracted, including the rationales tokens and shortcuts tokens (more examples are shown in Appendix B.1). Then, the shortcuts tokens will hurt the prediction performance. To alleviate this situation, we first construct a fine-grained causal graph (Figure 2(d)) between the selected rationale $R$ and the prediction results $Y$. Among them, $R$ represents the rationale generated by $\alpha$-constraint, $H$ denotes the context-specific representation of $R$ based on the context prior $C$. As mentioned before, from the graph, we find that as there exists a backdoor path $R \leftarrow C \rightarrow H \rightarrow Y$, $R$ and $Y$ are confounded. Then, based on the above observation, the *predictor* adopts the causal intervention methods described in section 3.2 (i.e., calculating $P(Y|do(R)) \approx \text{softmax}(\sum_{i=1}^{|N|} f_r(R, h_i)P(c_i))$ to remove the spurious correlations and further yield prediction results, where $f_r(\cdot)$ is the function to obtain $R \rightarrow Y \leftarrow H$, and $h_i = g_r(R, c_i)$ represents the process of $R \rightarrow H \leftarrow C$. Detailed descriptions of the graph at a high-level and the derivation are shown in Appendix A.2.

# 4 Experiments

In this section, we validate the effectiveness of our method on three real-world tasks including the beer reviews sentiment analysis, movies reviews prediction and the legal judgment prediction.

## 4.1 Beer Reviews Sentiment Analysis

Beer Reviews Sentiment Analysis aims to predict the ratings (at the scale of [0, 1]) for the multiple aspects of beer reviews (e.g., appearance, aroma and palate). We use the BeerAdvocate (McAuley et al., 2012) as our dataset, which is commonly used in the rationalization. As there is high sentiment correlation in different aspects in the same beer review (Lei et al., 2016), which may confuse the model training, several researches (Lei et al., 2016; Bastings et al., 2019) adopt the de-correlated sub-datasets (i.e., a part of BeerAdvocate) in the training stage. However, a high correlated dataset is more conducive to validating our Inter-RAT which is designed to remove the spurious correlations in data. Although (Chang et al., 2020) also conduct a correlated sub-dataset, the data split and processing are not available. For a fair comparison, different from the previous study which makes experiments on the sub-dataset, we train and validate models

on the original BeerAdvocate. Besides, following the setup of (Chang et al., 2020), we consider the beer review prediction as a binary classification where the ratings $\leq 0.4$ as *negative* and $\geq 0.6$ as *positive*. Then, the processed BeerAdvocate is a non-balanced dataset. For example, the label distribution in the appearance is *positive:negative* $\approx$ 20:1. For testing, we take manually annotated rationales as our test set, detailed statistics are shown in Appendix C.1.

### 4.1.1 Experimental Setup

In this section, we first present the comparison methods with Inter-RAT including RNP (Lei et al., 2016), HardKuma (Bastings et al., 2019), A2R (Yu et al., 2021), INVRAT (Chang et al., 2020), IB (Paranjape et al., 2020) and DARE (Yue et al., 2022). Among them, RNP is an original rationalization method which generates rationales by yielding the Bernoulli distribution of each token and sampling from it. HardKuma, A2R, IB and DARE all achieve promising results in rationalization. INVRAT is our main baseline to directly compare with, which is also a method of removing the spurious correlation in data. The difference with our Inter-RAT is that INVRAT learns environment invariant representations by obtaining multiple environments from the training set.

For training, we use the pre-trained glove embeddings (Pennington et al., 2014) with size 100, and implement the encoder in both $f_e(\cdot)$ and $f_p(\cdot)$ as the bidirectional GRU (Cho et al., 2014) with hidden size 100. We optimize the objective of rationalization using Adam (Kingma and Ba, 2014) with mini-batch size of 256 and an initial learning rate of 0.001. Besides, we consider the $\alpha$ in Eq (4) as {0.1, 0.2, 0.3}, respectively. For testing, we report the token precision, recall and F1-score to evaluate the quality of selected rationales. Among them, token precision is defined as the percentage of how many the selected tokens are in annotated rationales, and token recall is the percentage of annotated rationale tokens that are selected by model. The token F1-score is calculated as $\frac{2*\text{precision}*\text{recall}}{\text{precision}+\text{recall}}$.

### 4.1.2 Experimental Results

To demonstrate the effectiveness of our Inter-RAT, we briefly compare it with baselines (e.g. RNP and DARE) in Table 1, where Inter-RAT outperforms them consistently in finding correct rationales. Specifically, Inter-RAT surpasses baselines on all three aspects (i.e, appearance, aroma and

Table 1: Precision, Recall and F1 of selected rationales for the three aspect, where $\alpha$ is the predefined sparsity level.

| Methods | $\alpha$ | Appearance | | | Aroma | | | Palate | | |
|---|---|---|---|---|---|---|---|---|---|---|
| | | Precision | Recall | F1 | Precision | Recall | F1 | Precision | Recall | F1 |
| RNP | 0.1 | 32.4±0.5 | 18.6±0.3 | 23.6±0.4 | 44.8±0.4 | 32.4±0.7 | 37.6±0.5 | 24.6±0.5 | 23.5±0.5 | 24.0±0.5 |
| HardKuma | 0.1 | 53.6±0.1 | 28.7±0.1 | 37.4±0.1 | 29.3±1.4 | 25.9±3.8 | 27.3±2.1 | 7.7±0.1 | 6.0±0.1 | 6.8±0.1 |
| INVRAT | 0.1 | 42.6±0.7 | 31.5±0.6 | 36.2±0.6 | 41.2±0.3 | 39.1±2.8 | 40.1±1.6 | **34.9±1.5** | 45.6±0.2 | 39.5±1.0 |
| IB | 0.1 | 50.5±0.2 | 29.7±0.5 | 37.4±0.4 | 43.5±0.3 | 39.2±0.5 | 41.3±0.1 | 29.9±0.2 | 34.2±0.7 | 31.9±0.2 |
| DARE | 0.1 | 63.9±0.1 | 42.8±0.2 | 51.3±0.1 | 50.5±0.1 | 44.8±0.2 | 47.5±0.2 | 33.1±0.4 | 45.8±0.1 | 38.4±0.2 |
| Inter-RAT | 0.1 | **66.0±0.4** | **46.5±0.8** | **54.6±0.7** | **55.4±0.9** | **47.5±0.6** | **51.1±0.8** | 34.6±0.8 | **48.2±0.4** | **40.2±0.5** |
| RNP | 0.2 | 39.4±0.4 | 44.9±0.1 | 42.0±0.2 | 37.5±0.1 | 51.9±0.7 | 43.5±0.3 | 21.6±0.4 | 38.9±0.5 | 27.8±0.4 |
| HardKuma | 0.2 | **64.9±0.9** | 69.2±1.0 | 67.0±0.8 | 37.0±1.3 | 55.8±1.9 | 44.5±1.5 | 14.6±0.3 | 22.3±0.8 | 17.7±0.4 |
| INVRAT | 0.2 | 58.9±0.4 | 67.2±2.3 | 62.8±1.1 | 29.3±1.0 | 52.1±0.6 | 37.5±0.6 | 24.0±1.3 | 55.2±2.3 | 33.5±1.6 |
| IB | 0.2 | 59.3±0.4 | 69.0±0.2 | 63.8±0.2 | 38.6±0.1 | 55.5±0.7 | 45.6±0.1 | 21.6±0.2 | 48.5±0.4 | 29.9±0.2 |
| DARE | 0.2 | 63.7±0.2 | 71.8±0.8 | 67.5±0.5 | 41.0±0.2 | 61.5±0.2 | 49.3±0.3 | 24.4±0.1 | 54.9±0.8 | 33.8±0.1 |
| Inter-RAT | 0.2 | 62.0±0.5 | **76.7±1.7** | **68.6±0.4** | **44.2±0.1** | **65.4±0.2** | **52.8±0.1** | **26.3±0.6** | **59.1±0.8** | **36.4±0.7** |
| RNP | 0.3 | 24.2±0.4 | 41.2±0.8 | 30.5±0.5 | 27.1±0.3 | 55.7±0.8 | 36.4±0.4 | 15.4±0.4 | 42.2±0.9 | 22.6±0.5 |
| HardKuma | 0.3 | 42.1±0.3 | 82.4±1.4 | 55.7±0.5 | 24.6±0.1 | 57.7±0.6 | 34.5±0.2 | 21.7±0.1 | 49.7±0.4 | 30.2±0.1 |
| INVRAT | 0.3 | 41.5±0.4 | 74.8±0.3 | 53.4±0.3 | 22.8±1.6 | 65.1±1.7 | 33.8±1.8 | 20.9±1.1 | **71.6±0.4** | 32.3±1.3 |
| IB | 0.3 | 40.2±0.1 | 81.5±0.2 | 53.9±0.2 | 27.9±0.2 | 59.2±0.3 | 37.9±0.1 | 19.1±0.7 | 59.0±0.9 | 28.9±0.6 |
| DARE | 0.3 | 45.5±0.2 | 80.6±0.2 | 58.1±0.1 | 32.7±0.2 | 68.2±0.3 | 44.2±0.1 | 19.7±0.6 | 70.5±0.4 | 30.8±0.7 |
| Inter-RAT | 0.3 | **48.1±0.7** | **82.7±0.5** | **60.8±0.4** | **37.9±0.7** | **72.0±0.1** | **49.6±0.7** | **21.8±0.1** | 66.1±0.8 | **32.8±0.1** |

palate) by a large margin in most metrics. Besides, although INVRAT has shown helpful in discovering the de-confounded rationales, Inter-RAT still performs better than it, improving 10.5, 14.0 and 1.4 on the average token F1-score across three aspects, and Inter-RAT has a lower variance illustrating our method is more stable than INVRAT. Such observations strongly demonstrate that Inter-RAT can remove the spurious correlation in data to select rationales effectively.

As discussed in section 3.3, we propose the causal intervention method to alleviate the problem, where several tokens corresponding to spurious correlations in data may be selected and further mislead the prediction with an increasing $\alpha$. Here, we conduct an experiment to validate the effectiveness of the causal intervention. Since there is only about 1,000 beer reviews in the test set, we report the binary classification F1-score[3] with different $\alpha$ in the dev set which contains about 30,000 reviews. As shown in Figure 3(a), we make experiments on the palate aspect, and Inter-once is a variant of Inter-RAT, which yields the rationales based on $P(R|do(X))$ but predicts the results based on $P(Y|R)$, rather than $P(Y|do(R))$. From the observation, we can conclude that when $\alpha$ is small (i.e., the length of selected rationales is smaller than real rationales), the difference between Inter-RAT and Inter-once is minor. However, as $\alpha$ increases, Inter-RAT steadily improves, while the Inter-once grows slowly and even degrades. The above observation illustrates that our causal intervention method can alleviate the spurious correlations problem between $R$ and $Y$ caused by the $\alpha$-constraint.

As mentioned in section 4.1, we make experi-

---

[3]Different from token F1, F1-score is commonly adopted to evaluate the performance of binary classification.

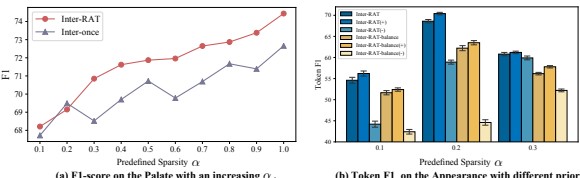

(a): F1-score on the Palate with an increasing $\alpha$.
(b): Token F1 on the Appearance with different prior.

Figure 3: (a): F1-score on the Palate with an increasing $\alpha$. (b): The token F1 for rationales on the appearance aspect with different prior distributions.

ments on the non-balanced dataset, which is different from the previous study (Chang et al., 2020, 2019; Huang et al., 2021) adopting the balanced datasets. Therefore, there exists a research question we need to answer:"Does the information of label distributions (or prior distributions) somehow influence Inter-RAT to yield better rationales instead of the causal intervention ?". For instance, as the label distribution in the appearance aspect is *positive:negative* $\approx$ 20:1, we conduct experiments on the appearance dataset with $P(c_1) = \frac{20}{21}$ and $P(c_2) = \frac{1}{21}$, where $P(c_i)$ in Eq (5) represents the prior distribution of $c_i$, $c_1$ is the *positive* label and $c_2$ is the *negative* one. The above non-balanced label distribution might be inducing the model to "better rationalize" for the majority class (i.e., the *positive*), further reflecting the improvement of Inter-RAT over the whole dataset. Therefore, we compute token-F1 scores for positive and negative examples separately for a safer evaluation, where we denote the evaluation of Inter-RAT on positive examples as Inter-RAT(+) and on the negative ones as Inter-RAT(-). Figure 3(b) summarizes the results on the appearance aspect. From the result, we find that the performance of extracting positive rationales is better than extracting the negative, although the difference between the two types results is not significant, and the scores for the negative are still high (better than INVRAT). Therefore, to

Table 2: Results on Movie, where several results of baselines are quoted from A2R (Yu et al., 2021).

| Methods | Movie | | |
|---|---|---|---|
| | Precision | Recall | F1 |
| RNP | 35.6 | 21.1 | 24.1 |
| Bert_RNP | 43.4 | 26.1 | 32.6 |
| HardKuma | 31.1 | 28.3 | 27.0 |
| A2R | **48.7** | 31.9 | 34.9 |
| INVRAT | 33.9 | 24.3 | 28.3 |
| Inter-RAT | 35.7±0.2 | 35.8±1.7 | 35.7±0.8 |
| Bert_Inter-RAT | 31.7±0.1 | **43.1±1.1** | **36.5±0.3** |

further validate the effect of label distribution, we add the analysis as follows: we re-run the experiments with $P(c_1) = P(c_2) = \frac{1}{2}$ (i.e., assuming this is a balanced dataset with uniform label distributions) and denote the corresponding model as Inter-RAT-balance. We report experimental results in Figure 3(b). From the figure, we find adopting the true prior distribution $P(c_i)$ (Inter-RAT) performs better than the assumed one (Inter-RAT-balance), demonstrating the prior distribution is critical for the backdoor adjustment. Besides, it is interesting to see that with a balanced label distribution, the results of the minority label (i.e., *negative*) are worse than using the true label distribution, which suggests Inter-RAT is not simply "paying more attention" to instances of the majority class.

## 4.2 Movies Reviews Prediction

Besides the beer reviews sentiment analysis task, we also make experiments on another binary classification task (i.e., movie review prediction (Zaidan and Eisner, 2008)) in the ERASER benchmark (DeYoung et al., 2020), which contains token-level human annotations. We follow the same experimental setups in 4.1.1 and report the experimental results with $\alpha = 0.2$ in Table 2. Detailed description of the dataset is shown in Appendix C.1. As shown in the table, Inter-RAT performs better than baselines on most metrics, which further validates the effectiveness of Inter-RAT. Furthermore, to validate Inter-RAT is agnostic to the structure of the *selector* and *predictor*, we adopt Bert (Devlin et al., 2019) to replace bi-GRU in $f_e(\cdot)$ and $f_p(\cdot)$ in both RNP and Inter-RAT, and denote them as Bert_RNP and Bert_Inter-RAT. From the result, we observe that Bert_Inter-RAT still outperforms Bert_RNP, illustrating the effectiveness of Inter-RAT.

## 4.3 Legal Judgment Prediction

Since there are only two categories (*positive* and *negative*) in both beer and movie reviews prediction, we further generalize our Inter-RAT to the multi-classification task. Specifically, we focus on the Legal Judgment Prediction (LJP) task, which yields the judgment results such as the charges based on the case fact. We conduct experiments on the publicly LJP datasets CAIL (Xiao et al., 2018) which contains criminal cases consisting of the fact description and corresponding charges, law articles, and terms of penalty results. For data processing, referring to (Yue et al., 2021), we remove several infrequent and multiple charges cases, and divide the terms into non-overlapping intervals. The detailed statistics of the datasets can be found in (Yue et al., 2021). Figure 1 shows an example of LJP, which predicts the charge according to the case fact.

### 4.3.1 Experimental Setup

In addition to comparing RNP, HardKuma and INVRAT, we also compare our method with some classical baselines in the LJP task, including Top-Judge (Zhong et al., 2018), Few-Shot (Hu et al., 2018), LADAN (Xu et al., 2020) and NeurJudge (Yue et al., 2021). All the above baselines are trained by exploiting legal particularities. where NeurJudge is the state-of-the-art model in LJP. Meanwhile, it employs a label embedding method to enhance the prediction. We conduct experiments on one of versions of CAIL containing 134,739 cases (Yue et al., 2021). For testing, as there are no annotated rationales, we first employ the accuracy (Acc), macro-precision (MP), macro-recall (MR), and macro-F1 (F1) to evaluate the performance of yielding judgment results. Then, we provide a human evaluation for selected rationales in LJP. Detailed description of comparison methods and experimental setups can be found in Appendix C.2.

### 4.3.2 Experimental Results

To evaluate the performance of our model on LJP, we show the experimental results from two aspects. First, Table 3 shows that our Inter-RAT still performs better than the rationalization methods when generalizing to the multi-classification task. Meanwhile, compared with the LJP approaches (e.g. Top-Judge and NeurJudge), even though our model is trained on the three subtasks separately, while these LJP approaches explore the dependencies between tasks and are trained with a multi-task learning framework, our model still achieves promising performance. However, Inter-RAT does not perform better than NeurJudge. A potential reason is that NeurJudge is designed only for LJP, exploiting the legal particularities well. In contract, our Inter-RAT is designed for general text classification tasks.

Table 3: LJP results on CAIL. Among them, the underline scores are the state-of-the-art performances in LJP but lacking explainability, and the results in **bold** perform second only to NeurJudge but with an explainable rationale. Results of LJP baselines are quoted from (Yue et al., 2021).

| Methods | Charges | | | | Law Articles | | | | Terms of Penalty | | | |
|---|---|---|---|---|---|---|---|---|---|---|---|---|
| | Acc | MP | MR | F1 | Acc | MP | MR | F1 | Acc | MP | MR | F1 |
| TopJudge | 86.5 | 84.2 | 78.4 | 80.2 | 87.3 | 85.8 | 76.3 | 78.2 | 38.4 | 35.7 | 32.2 | 31.3 |
| Few-Shot | 88.2 | 87.5 | 80.6 | 82.0 | 88.4 | 86.8 | 77.9 | 79.5 | 39.6 | 37.1 | 30.9 | 31.6 |
| LADAN | 88.3 | 86.4 | 80.5 | 82.1 | 88.8 | 85.2 | 79.5 | 81.0 | 38.1 | 34.0 | 31.2 | 30.2 |
| NeurJudge | 89.9 | 87.8 | 86.8 | 87.0 | 90.4 | 87.2 | 85.8 | 86.1 | 41.7 | 40.4 | 37.2 | 37.3 |
| RNP | 85.1±0.2 | 82.2±0.3 | 78.1±0.5 | 79.0±0.5 | 86.5±0.1 | 82.1±0.3 | 77.7±0.7 | 78.8±0.6 | 37.1±0.2 | 30.1±0.3 | 30.3±0.4 | 27.7±0.3 |
| HardKuma | 86.2±0.2 | 84.7±1.2 | 79.0±0.8 | 80.6±0.4 | 86.8±0.8 | 83.3±1.7 | 77.3±0.8 | 78.9±0.9 | 35.8±0.1 | 34.3±1.3 | 27.3±0.3 | 25.8±0.4 |
| INVRAT | 85.4±0.2 | 83.3±0.1 | 78.7±0.3 | 80.2±0.3 | 85.1±0.1 | 83.1±0.1 | 76.2±0.1 | 78.1±0.2 | 38.5±0.3 | 34.5±0.7 | 33.0±0.2 | 32.0±0.5 |
| Inter-RAT | **89.4±0.2** | **87.5±0.3** | **85.3±0.3** | **85.9±0.3** | **89.5±0.1** | **86.3±0.1** | **83.5±0.3** | **84.6±0.3** | **39.6±0.1** | **36.3±0.4** | **34.3±0.3** | **32.8±0.3** |

Table 4: Human evaluation on charge prediction.

| Methods | U | C | F | Avg. |
|---|---|---|---|---|
| RNP | 3.85 | 3.28 | 3.34 | 3.49 |
| INVRAT | 3.88 | 3.42 | 3.41 | 3.57 |
| HardKuma | 3.78 | 3.33 | 3.39 | 3.50 |
| DARE | **4.57** | 3.89 | 4.19 | 4.22 |
| ChatGPT | 4.01 | 4.08 | **4.34** | 4.14 |
| Inter-RAT | 4.52 | **4.10** | 4.25 | **4.29** |

Therefore, the performance of Inter-RAT does not surpass NeurJudge. Furthermore, different from the NeurJudge and other LJP baselines, our Inter-RAT can provide an intuitive explanation (i.e., rationales) when yielding the judgment results while LJP baselines fail to produce them. The above observation provides strong validation of adopting the causal intervention method to remove spurious correlation in data for predicting results. Interestingly, we find there exists a minor difference between Inter-RAT and NeurJudge on yielding the charge and law article. We argue a potential reason is the label embedding method in NeurJudge can be approximated as the causal intervention method. We further discuss it in Appendix D.2 in detail.

Second, as CAIL does not provide annotated rationales like BeerAdvocate, we make a human evaluation to evaluate the selected rationales. Specifically, we sample 100 examples and ask three annotators who are both good at computer science and law to evaluate rationales in the charge prediction. Besides, following (Sha et al., 2021), we employ three metrics with an interval from 1 (lowest) to 5 (e.g. 2.0 and 3.2) to evaluate rationales, including usefulness (U), completeness (C), and fluency (F). Appendix C.3 describes detailed scoring standards for human annotators. The human evaluation results are shown in Table 4. It is worth noting that a similar human evaluation is provided in DARE, but its is set differently from ours, where DARE sets $\alpha$ to 0.14 and our Inter-RAT is set to 0.2. For consistency, we replicate DARE with $\alpha = 0.2$.

From the results, we can find Inter-RAT outperforms RNP, INVRAT, HardKuma and DARE in all metrics, further demonstrating our causal intervention method can select more sufficient rationales for yielding results. Besides, considering the success of Large Language Models (LLMs), we additionally add ChatGPT (OpenAI, 2023) for human evaluation (the prompt for ChatGPT is shown in Appendix C.4). As shown in Table 4, we observe that ChatGPT achieves competitive results on both Completeness and Fluency. But it does not perform well on Usefulness. Because even though we ask ChatGPT to extract short tokens and sentences as rationale, it still tends to extract longer sentences, which is beneficial for the Fluency metric. However, for the Usefulness metric, some redundant tokens will be extracted as well. Therefore, Chat-GPT does not perform well on the Usefulness.

## 5 Related Work

**Rationalization.** Deep neural networks (DNNs) have achieved remarkable success in various domains (Seo et al., 2016; Devlin et al., 2019; Liu et al., 2023c; Gao et al., 2023). However, the predicted results are still unreliable. To improve the explainability of DNNs, the rationalization has attracted increasing attention (Lei et al., 2016; Treviso and Martins, 2020; Bastings et al., 2019; Chang et al., 2019; Liu et al., 2023a,b). Specifically, (Lei et al., 2016) first proposed a rationalization framework which consists of a *selector* and a *predictor*. Following this framework, multiple variants were proposed to improve rationalization. Among them, to replace the Bernoulli sampling distribution in (Lei et al., 2016), (Bastings et al., 2019) introduced a HardKuma distribution for reparameterized gradient estimates. Additionally, another fundamental direction is adding external components to enhance the original framework. (Yu et al., 2019) employed an introspective *selector* which in-

corporated the prediction results into the selection process. Some researchers (Huang et al., 2021; Sha et al., 2021; Cao et al., 2020) proposed an external guider to reduce the difference between the distributions of rationales and input. However, few considered spurious correlations in data which degraded the rationalization. Among them, (Chang et al., 2020) discovered the causal rationales with environment invariant methods by creating different environments. (Wu et al., 2022) extracted rationales from graphs to study the explainability of graph neural networks by intervention distributions.

**Causal Inference.** Causal inference (Glymour et al., 2016) has been widely explored in various fields, including medicine (Richiardi et al., 2013) and politics (Keele, 2015), which aims to empower models the ability to achieve the causal effect. Recently, several researches (Deng and Zhang, 2021; Dong et al., 2020; Yue et al., 2020; Niu et al., 2021; Vosoughi et al., 2023) introduced causal inference into machine learning with causal intervention to remove the spurious correlations in data. Especially, it has inspired several studies in natural language understanding such as Named Entity Recognition (Zhang et al., 2021), Topic modeling (Wu et al., 2021), and Entity Bias problem (Zhu et al., 2022; Wang et al., 2023).In this paper, we focus on improving the rationalization with causal intervention.

## 6 Conclusion

In this paper, we proposed a causal intervention method (Inter-RAT) to improve rationalization. To be specific, we first formulated the causalities in rationalization with a causal graph and revealed how the confounder hurt the performance of selecting rationales with opened backdoor paths. Then, we introduced a backdoor adjustment method to remove spurious correlations between inputs and rationales. Besides, we further discussed the potential bias between selected rationales and predicted results caused by the sparsity constraints, and adopted the above causal intervention method to yield de-confounded prediction results. Experimental results on real-world datasets have clearly demonstrated the effectiveness of Inter-RAT.

## Limitations

To achieve the causal intervention, we adopt the backdoor adjustment to calculate $P(R|do(X))$ and $P(Y|do(R))$ with the observed label as the confounder set. However, in some cases, observable labels are still difficult to obtain. For example, if we take the beer review prediction as a text regression task like (Bastings et al., 2019), rather than a classification task, then we will fail to obtain the observed label, where the ratings of beer reviews are continuous. Therefore, to achieve the causal intervention, we argue that a promising approach is to adopt other intervention methods, such as the instrumental variable (Yuan et al., 2022; Singh et al., 2019) that is implemented with the unobserved confounder. We will leave these for the future work. Besides, since our approach focuses on how to improve the explainability of neural networks and is a model-agnostic approach, we will also study in the future how to employ rationalization methods to improve the Large Language Models (LLMs), considering the recent remarkable success of LLMs (OpenAI, 2023; Touvron et al., 2023).

## Acknowledgements

This research was supported by grant from the National Key Research and Development Program of China (Grant No. 2021YFF0901003).

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

## A Instantiated Causal Graph

### A.1 Causal graph for the selector

In this section, we describe causal graph for the *selector* (Figure 2(b)) in detail with examples:

$C \to X$. The context prior $C$ determines which tokens can be "put" into the text input $X$. Among them, the context prior consists of unobserved prior and observed prior (such as the label set). For example, in Figure 1, both observed *Intentional homicide* and *Manslaughter* priors decide where the tokens denoting violence and death appear ; the *Manslaughter* prior determines where the tokens representing manslaughter appear ; the unobserved prior decides where other tokens that are meaningless appear.

$X \to K \leftarrow C$. $K$ denotes the context-specific representation which is a weighted representation of the prior knowledge associated with $X$ in $C$. Taking Figure 1 as an example, we assume that the label set consisting of *Intentional homicide*, *Manslaughter*, and *Theft* is the observed prior. Then, we can get the context prior which consists of four parts (i.e., the *Intentional homicide* prior $c_1$, the *Manslaughter* prior $c_2$, the *Theft* prior $c_3$ and the unobserved prior $c_4$). Next, we calculate the association between $X$ and $C$, and obtain the corresponding scores, assuming $a_1 = 0.3$, $a_2 = 0.6$, $a_3 = 0.0$, $a_4 = 0.1$, where *Manslaughter* prior $c_2$ and $X$ are the most relevant, and the *Theft* prior $c_3$ and $X$ are the least relevant. Finally, we can calculate the $K$ as $a_1c_1 + a_2c_2 + a_3c_3 + a_4c_4 = 0.3c_1 + 0.6c_2 + 0.1c_4$.

$X \to R \leftarrow K$. As the rationale $R$ is a subsequence of $X$, $X \to R$ holds. Besides, $K \to R$ represents the contextual constitution of the text that affects the composition of rationales. Taking the previous example as an example, since $K$ is calculated as $0.3c_1 + 0.6c_2 + 0.1c_4$, tokens in $R$ will be more inclined with the *Manslaughter* prior $c_2$.

### A.2 Causal graph for the predictor

In this section, we describe the detailed causal relationship (Figure 4(a)) between the selected rationales $R$ and the results $Y$ in the *predictor* at a high-level:

$C \to R$. Based on Eq (5) in section 3.2, we can conclude that the context prior $C$ determines which tokens are corresponding to rationales.

$R \to H \leftarrow C$. $H$ represents the context-specific representation of $R$ by using the context prior $C$. $R \to Y \leftarrow H$. As rationales $R$ are selected by $\alpha$-

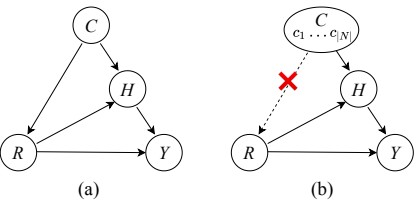

Figure 4: Causal graph for the *predictor*.

constraint and consist of real rationales and short-cuts tokens, we argue that $R \to Y$ holds. Besides, the context prior $C$ affects the label $Y$ by the mediator $H$. The reason is similar to $K \to R$ in Appendix A.1.

From the graph, we can clearly see the context prior $C$ is the confounder between $R$ and $Y$, which opens the backdoor path $R \leftarrow C \to H \to Y$. Therefore, to remove the spurious correlations between $R$ and $Y$, we adopt the causal intervention method to calculate $P(Y|do(R))$ by cutting the link $C \to R$ (Figure 4(b)):

$$
\begin{aligned}
P(Y|do(R)) &= \sum_{i=1}^{|N|} \left[ P\left(Y|R, H\right) P\left(c_i\right) \right] \\
&= \sum_{i=1}^{|N|} \left[ P\left(Y|f_r(R, h_i)\right) P\left(c_i\right) \right] \\
&= \sum_{i=1}^{|N|} \left[ \text{softmax}(f_r(R, h_i)) P\left(c_i\right) \right],
\end{aligned}
\tag{8}
$$

where $f_r(\cdot)$ is the function achieving $R \to Y \leftarrow H$, $h_i = g_r(R, c_i) = \beta_i c_i$, $\beta_i \in \beta$. $\beta \in \mathbb{R}^N$ is the set of the normalized similarity between $R$ and each $c_i$ in the confounder set $C$ (i.e., $\beta = \text{softmax}(f_p(R)D_c^T)$). Among them, $f_p(\cdot)$ encodes $R$ into a d-dimensional vector, and the dictionary $D_c = \{c_1, c_2, \ldots, c_{|N|}\}$ is approximated as the observed confounder.

Besides, we also adopt the NWGM approximation to Eq (8) and set $f_r(R, h_i)$ as a linear model (i.e., $f_r(R, h_i) = W_3 f_p(R) + W_4 h_i = W_3 f_p(R) + W_4 \beta_i c_i$). Then, the final objective of intervention is formulated as:

$$
\begin{aligned}
P(Y|do(R)) &= \sum_{i=1}^{|N|} \left[ \text{softmax}(f_r(R, h_i)) P\left(c_i\right) \right] \\
&\approx \text{softmax}(\sum_{i=1}^{|N|} f_r(R, h_i) P\left(c_i\right)) \\
&= \text{softmax}(W_3 f_p(R) \\
&\quad + W_4 \sum_{i=1}^{|N|} \beta_i c_i P\left(c_i\right)).
\end{aligned}
\tag{9}
$$

Table 5: Detailed statistics of the processed BeerAdvocate and MovieReview.

| Dataset | Train | | | Dev | | | Test | | | |
|---|---|---|---|---|---|---|---|---|---|---|
| | # Pos | # Neg | # Avg.Tokens | # Pos | # Neg | # Avg.Tokens | # Pos | # Neg | # Avg.Tokens | # Avg.Rationale |
| BeerAdvocate (appearance) | 202,385 | 12,897 | 153.86 | 28,488 | 1,318 | 151.33 | 955 | 14 | 126.80 | 22.61 |
| BeerAdvocate (aroma) | 172,299 | 30,564 | 154.80 | 24,494 | 3,396 | 152.46 | 913 | 31 | 126.62 | 18.38 |
| BeerAdvocate (palate) | 176,038 | 27,639 | 153.98 | 24,837 | 3,203 | 151.77 | 940 | 24 | 126.02 | 13.46 |
| MovieReview | 800 | 800 | 847.25 | 100 | 100 | 835.03 | 99 | 100 | 835.03 | 54.40 |

### A.3 Why there is no need to perform interventions on the front-door path?

In this paper, we only block the backdoor pathways in Figure 2, unable to eliminate front-door pathway. The reason why we do not perform interventions on the front-door path is as follows:

1. Firstly, from the causal graph, we find the backdoor path $X \leftarrow C \rightarrow K \rightarrow R$ causes the spurious correlation. Meanwhile, the increased likelihood of $R$ given $X$ is due to "$X$ causes $R$" via $X \rightarrow R$ and $X \rightarrow K \rightarrow R$ (the mediation path).

2. In $X \rightarrow K \rightarrow R$, $K$ is a projection of $X$ in the context prior $C$. Therefore, we argue that the $X \rightarrow R$ path can be removed if $X$ can be fully represented by $K$, where $X$ also requires a neural network representation to achieve $X \rightarrow R$. Therefore, $X \rightarrow K \rightarrow R$ is beneficial for prediction.

3. In our paper, we aim to use the do-operation which implements the de-confounded training of the rationalization to remove the spurious correlation. Meanwhile, we retain the mediation path to keep the beneficial things for prediction.

## B Explanations for $\alpha$-constraint

### B.1 Examples for $\alpha$-constraint

As mentioned in section 3.3, in the practical, a higher predefined sparsity level $\alpha$ may bring shortcuts tokens which hurt the prediction performance, an extreme example being that all tokens in the text input will be selected. Below, we take a beer review as an example to further illustrate this problem, where this example is adopted to predict scores of the smell aspect.

**The original text:**
*He thinks this beer smells great and tastes terrific .*
**Rationales:** *smells great*
**Rationales with shortcuts tokens**, where $\alpha$ is set to 0.5: *smells great and tastes terrific*. Among them, *"and tastes terrific"* can be considered as shortcuts tokens.

### B.2 Discussions on $\alpha$-constraint

Although several rationalization methods do not set $\alpha$-constraint to extract rationales, we believe that their methods of constraining the short rationales extraction can be considered a variant of $\alpha$-constraint, and our intervention method in section 3.3 will still be effective on these methods. Specifically, we argue that these methods should set hyperparameters to encourage the model to select short rationales. However, if the hyperparameters are not set properly, it is possible that more shortcuts tokens will be extracted, making $R$ and $Y$ confounded. For example, for several methods (Chen and Ji, 2020; Paranjape et al., 2020) adopting the information bottleneck to ensure the model extracts short rationales, there exists a KL divergence between the posterior distribution $P(\widetilde{m}_j|x_j)$ and the prior distribution $r(\widetilde{m}_j)$, where $r(\widetilde{m}_j) =$ Bernoulli$(\pi)$ for some constant $\pi \in (0, 1)$. For instance, if we set $\pi$ as $0.1$, it means we encourage the model to extract 10% of the input text. Therefore, we consider $\pi$ as a variant of $\alpha$ proposed by us.

## C Setting Details

### C.1 Statistics of BeerAdvocate and MovieReview

In this section, we show the detailed statistics of BeerAdvocate and MovieReview in Table 5. Among them, BeerAdvocate contains three aspects, including appearance, aroma and palate. From the Table 5, we can observe that the processed BeerAdvocate is a non-balanced dataset. In the training set, the prior distribution is *positive:negative* $\approx 20$:1 in appearance, *positive:negative* $\approx 17$:3 in aroma, *positive:negative* $\approx 17$:3 in palate. Meanwhile, MovieReview is a balanced dataset with *positive:negative* $= 1$:1.

### C.2 Comparison methods and experimental setups for LJP

In addition to comparing RNP (Lei et al., 2016), HardKuma (Bastings et al., 2019) and INVRAT

(Chang et al., 2020), we also compare our method with some classical baselines in the LJP task:

- **TopJudge** (Zhong et al., 2018) explores the dependencies among the three subtasks in LJP.

- **Few-Shot** (Hu et al., 2018) utilizes the charge attributes to identify the confusing charges.

- **LADAN** (Xu et al., 2020) learns the distinguished law articles representations for LJP prediction.

- **NeurJudge** (Yue et al., 2021) is a circumstance aware approach adopting different crime circumstances to yield corresponding results. Meanwhile, it employs a label embedding method to enhance the prediction.

For training, we adopt the word2vec (Mikolov et al., 2013) for word embedding pre-training with size 200, and set the encoder in $f_e(\cdot)$ and $f_p(\cdot)$ as Bi-GRU. Besides, we implement the learning rate of 0.0002 with batch size 256, and take $\alpha$ as 0.2. For evaluating, we employ the accuracy (Acc), macro-precision (MP), macro-recall (MR), and macro-F1 (F1) to evaluate the performance of yielding judgment results.

## C.3 Scoring standards for human evaluation

Following (Sha et al., 2021), we evaluate the rationales with three metrics: usefulness (U), completeness (C), and fluency (F) in the charge prediction. Among them, each scored from 1 (lowest) to 5. Below, we introduce scoring standards for the above metrics in brief. Detailed standards for human annotators can be found in (Sha et al., 2021).

### C.3.1 Usefulness

Q: Do you think the selected rationales can be useful for explaining the predicted labels?

- 5: Exactly. Selected rationales are useful for me to get the correct label.

- 4: Highly useful. Although several tokens have no relevance to correct label, most selected tokens are useful to explain the labels.

- 3: Half of them are useful. About half of the tokens are useful for getting labels.

- 2: Almost useless. Almost all of the tokens are useless.

- 1: No Use. The selected rationales are useless for identifying labels.

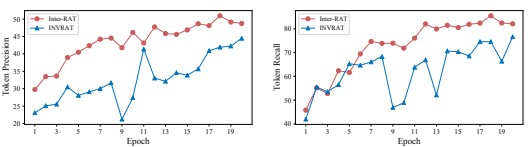

Figure 5: The token precision and recall for rationales on the appearance aspect with $\alpha = 0.3$.

### C.3.2 Completeness

Q: Do you think the selected rationales are enough for explaining the predicted labels?

- 5: Exactly. Selected rationales are enough for me to get the correct label.

- 4: Highly complete. Several tokens related to the label are missing.

- 3: Half complete. There are still some important tokens that have not been selected, and they are in nearly the same number as the selected tokens.

- 2: Somewhat complete. The selected tokens are not enough.

- 1: Nonsense. None of the important tokens is selected.

### C.3.3 Fluency

Q: Do you think the selected rationales are fluent?

- 5: Very fluent.

- 4: Highly fluent.

- 3: Partial fluent.

- 2: Very unfluent.

- 1: Nonsense.

## C.4 Prompt for ChatGPT for human evaluation

This is our prompt for ChatGPT:

Suppose you are a data tagging assistant and I will provide you with some case facts. Please follow the following prompts to make your predictions:

1. Please predict the charge of the given case facts.

2. Please extract one or more consecutive sentences and tokens from the case facts to support your prediction and keep the extracted sentences and tokens as short as possible. Please note that you must only extract from the given case facts.

3. Only output a python dictionary format to me, e.g. {'label':intentional homicide, 'rationale':extract sentence}.

The defendant and the victim were both students, after the dormitory relocation, in the new dormitory, the defendant and the victim for sleeping in the lower bunk bed dispute, **the defendant picked up a bottle and forcefully smashed the victim's head, resulting in the victim's head injury**, then the defendant sent the victim to hospital. **The victim died in hospital treatment failed.** The forensic medical appraisal showed the injury to the victim's face was a pre-existing injury and the degree of injury was minor. The victim's death was consistent with an acute heart attack where trauma and emotional stress as precipitating factors…

The defendant and the victim were both students, after the dormitory relocation, in the new dormitory, the defendant and the victim **for sleeping in the lower bunk bed dispute**, **the defendant picked up a bottle and** forcefully **smashed the victim's head, resulting in** the victim's **head injury**, then the defendant **sent the victim to hospital**. **The victim died** in hospital treatment failed. The forensic medical appraisal showed the injury to the victim's face was a pre-existing injury and the degree of injury was minor. The victim's death was consistent with an acute heart attack where trauma and emotional stress as precipitating factors…

The defendant and the victim were both students, after the dormitory relocation, in the new dormitory, the defendant and the victim **for sleeping in the lower bunk bed dispute**, **the defendant picked up a bottle and forcefully smashed the victim's head**, resulting in the victim's **head injury**, then **the defendant sent the victim to hospital**. **The victim died** in hospital treatment failed. The forensic medical appraisal showed the injury to the victim's face was a pre-existing injury and the degree of **injury was minor**. **The victim's death was consistent with an acute heart attack** where trauma and emotional stress as precipitating factors…

Figure 6: Examples of selective rationalization on the charge prediction. Among them, **RNP** wrongly predicts the charge as *Intentional homicide*. Meanwhile, both **INVRAT** and **Inter-RAT** predict the charge correctly, but Inter-RAT extracts more plausible rationales to yield results.

# D    More Experimental Results

## D.1    Changes with Training Epochs

Besides, comparing with INVRAT, we investigate the model performance by showing the changes in token precision and recall with training epochs. Figure 5 shows the experiments on the appearance aspect with $\alpha = 0.3$. From the observation, we can conclude that Inter-RAT significantly outperforms INVRAT in both precision and recall with lower variance from the training onwards, which proves the effectiveness of our proposed method.

## D.2    A Causal View on NeurJudge

In this section, from the causal view, we discuss the reason why the difference between Inter-RAT and NeurJudge on the charge and law article prediction is not significant. Here, we explain the observation by taking the charge prediction as an example, and the article prediction is similar. Specifically, NeurJudge adopts a label embedding method to incorporate the semantics of charge into the case fact to yield the corresponding result. We argue that this method can be approximated as the causal intervention method. To illustrate this discovery, we assume Figure 2(d) is the causal graph of the charge prediction task, and consider the case fact as $R$ (i.e., $\alpha$=1) and the charge label set as $C$. Then the process of label embedding can be formulated as $R \to H \leftarrow C$ and $R \to Y \leftarrow H$. The objective of NeurJudge is written as:

$$P(Y|R) = \text{softmax}(\sum_{i=1}^{|N|} f_{neru}(R, h_i)), \quad (10)$$

where $h_i = g_{neru}(R, c_i)$. We can find the difference between the Eq (10) and our causal interven-

tion method is that Eq (10) ignores the prior distribution $P(c_i)$. It is worth noting that although NeurJudge ignores $P(c_i)$, it performs slightly better than our model. A potential reason is that NeurJudge exploits the dependencies among LJP tasks well, while our model is trained on the independent task.

## D.3    Visualization

We provide several visualization cases in the CAIL dataset as shown in Figure 6, which are selected by RNP, Inter-RAT and INVRAT. Among them, annotated rationales are underlined. RNP, INVRAT and Inter-RAT rationales are highlighted in **yellow**, **green** and **pink** colors, respectively. From the Figure 6, we can find that RNP fails to predict the charge and considers it as *Intentional homicide* with capturing the tokens denoting violence and death. On the contrary, both Inter-RAT and INVRAT predict the charge correctly, but Inter-RAT can extract more comprehensive rationales (i.e., *The victim's death was consistent with an acute heart attack*), which support the victim's death was due to the negligence. This observation further demonstrates our causal intervention method can select more sufficient rationales for yielding results.