# OpenReview forum: "Interventional Rationalization"
_EMNLP/2023/Conference — EMNLP 2023 Main_

### Official Review · Reviewer_gxKG · 2023-08-03

**Soundness:** 3

**Excitement:**

4: Strong: This paper deepens the understanding of some phenomenon or lowers the barriers to an existing research direction.

**Paper Topic And Main Contributions:**

This paper introduces interventional rationalization as a novel method that leverages causal theory and causal intervention techniques to enhance the generation of explanations for neural network predictions. It tackles the challenge of spurious correlations and validates its effectiveness through experiments on real-world datasets.

**Reasons To Accept:**

 Using the structural causal model backdoor adjustment method to remove spurious correlations between inputs and rationales for text prediction tasks is meaningful for downstream analysis.

**Reasons To Reject:**

1. The causal graph in your Interventional Rationalization is similar to [1]. Both of you utilize causal intervention via backdoor adjustment and regard the category as the confounder. The causal graph model of the paper is fundamentally inspired by [1].

2. The effectiveness of using the text category as the confounder is debatable.

3. ''the underlined tokens in Figure 1 denoting the offense is negligent will be ignored in rationales extraction and the charge will be misjudged''. So the reason why the charge will be misjudged is because of the Entity Bias problem. The paper should introduce more about the works of Entity Bias.

[1] Zhongqi Yue, Hanwang Zhang, Qianru Sun, and Xian Sheng Hua. 2020. Interventional few-shot learning. In NeurIPS.

**Reproducibility:**

4: Could mostly reproduce the results, but there may be some variation because of sample variance or minor variations in their interpretation of the protocol or method.

**Reviewer Confidence:**

4: Quite sure. I tried to check the important points carefully. It's unlikely, though conceivable, that I missed something that should affect my ratings.

---

> ### Author Rebuttal · Authors · 2023-08-28
>
> Thank you for your time and insightful suggestions! According to your comments, we provide the responses as follows:
>
> > **Comment 1:** The paper should introduce more about the works of Entity Bias.
>
> Thanks for your comments, below we present some related work on Entity Bias:
>
> The Entity Bias problem is related to the spurious correlation problem in the rationalization.
>
> The Entity bias is a phenomenon that occurs when models overly rely on prediction shortcuts triggered by certain entities and thus makes spurious predictions.
>
> Numerous research works [1-5] have studied this. Among them, [1] proposes  an entity-masked contrastive pre-training framework for relation extraction to model the textual context and entity types.
>
> [4] proposes an entity debiasing framework by mitigating entity bias from a cause-effect perspective.
>
> [5] studies the causal intervention techniques to mitigate entity bias in language models.
>
> Different from the Entity Bias problem, rationalization is more concerned with removing spurious correlations in the data to improve the explainability of the model.
>
> Please let us know if there is any key related work that has been missed, and thanks again for your comments.
>
>
>
> > **Comment 2:** The effectiveness of using the text category as the confounder is debatable.
>
> Based on the existing research [6,7], we consider the label set as the partially observed children of the unobserved confounder $C$. Therefore, we approximate it by designing a dictionary $D_{c} = \{c_{1}, c_{2},  \ldots, c_{|N|}\}$ as an $N \times d$ matrix, where N is the number of labels.
>
> However, using text category as the confounder is an expedient measure, and we need more observable confounders. Inspired by [8], we can introduce fine-grained attributes of text category into consideration and take these attributes as additional confounders. We will make it our future work.
>
>
>
> > **Comment 3:** The causal graph model of the paper is fundamentally inspired by [7].
>
> Our causal graph model is inspired by [7]. Differently, in the rationalization, in addition to the spurious correlations between the textual input and  the extracted rationale, we futher focus on the limitations on the predeﬁned sparsity, where the sparsity constraint may result in spurious correlations between the extracted rationale and the predicted result.
>
> **References**
>
> [1] Hao Peng, Tianyu Gao, Xu Han, Yankai Lin, Peng Li, Zhiyuan Liu, Maosong Sun, and Jie Zhou. Learning from context or names? an empirical study on neural relation extraction. In EMNLP2020.
>
> [2] Shayne Longpre, Kartik Perisetla, Anthony Chen, Nikhil Ramesh, Chris DuBois, and Sameer Singh. Entity-based knowledge conflicts in question answering. In EMNLP2021.
>
> [3] Nan Xu, Fei Wang, Bangzheng Li, Mingtao Dong, Muhao Chen. Does Your Model Classify Entities Reasonably? Diagnosing and Mitigating Spurious Correlations in Entity Typing. In EMNLP2022.
>
> [4] Yongchun Zhu, Qiang Sheng, Juan Cao, Shuokai Li, Danding Wang, Fuzhen Zhuang. Generalizing to the Future: Mitigating Entity Bias in Fake News Detection. In SIGIR2022.
>
> [5] Fei Wang, Wenjie Mo, Yiwei Wang, Wenxuan Zhou, and Muhao Chen. A Causal View of Entity Bias in (Large) Language Models. In Arixv2023.
>
> [6] Tan Wang, Jianqiang Huang,Hanwang Zhang, Qianru Sun. Visual commonsense r-cnn. In CVPR2020.
>
> [7] Zhongqi Yue, Hanwang Zhang, Qianru Sun, and Xian Sheng Hua.  Interventional few-shot learning. In NeurIPS2020.
>
> [8] Zikun Hu, Xiang Li, Cunchao Tu, Zhiyuan Liu, and Maosong Sun. Few-shot charge prediction with discriminative legal attributes. In COLING2018.

---

### Official Review · Reviewer_sFn6 · 2023-08-09

**Soundness:** 3

**Excitement:**

3: Ambivalent: It has merits (e.g., it reports state-of-the-art results, the idea is nice), but there are key weaknesses (e.g., it describes incremental work), and it can significantly benefit from another round of revision. However, I won't object to accepting it if my co-reviewers champion it.

**Missing References:**

above

**Paper Topic And Main Contributions:**

This paper aims to analyze from a causal relationship perspective why models for Selective Rationalizations tasks are prone to learning shortcuts rather than true causal patterns, for the purpose of enhancing explainability. It also simulates the implementation of the 'do' operator in deep learning, which has shown commendable performance on multiple datasets.

**Questions For The Authors:**

1、Could you provide statistical results relevant to the shortcut mentioned in Figure 1? I couldn't accept that there is only a narrative ``story'' claiming that the issue is caused by the shortcut. After all, whether introducing spurious correlations or identifying the shortcut, we require obvious experimental results for validation.

2、There are indeed some works that analyze the causal relationships between shortcuts, rationales, and prediction outcomes. Why not elaborate the differences with these works in the related work section? For instance, the referenced paper [1] and [2] proposed a causal graph that bears no essential difference from the one in this paper, and it's even clearer.

[1]:Ying-Xin Wu, Xiang Wang, An Zhang, Xiangnan He, and Tat seng Chua. 2022. Discovering invariant rationales for graph neural networks. In ICLR.
[2]:Fan, Shaohua, et al. "Debiasing graph neural networks via learning disentangled causal substructure." Advances in Neural Information Processing Systems 35 (2022): 24934-24946.

**Reasons To Accept:**

1. The performance shown in the experiments is well.
2. The imposition of sparsity constraints is a good insight.

**Reasons To Reject:**

1. Figure 2 is not an SCM  structure but a simple DAG (Directed Acyclic Graph) format. An SCM structure includes not only the value of nodes and edges but also the important aspect of independent noise terms, which allows judgment of the causal relationship between pairs of variables. However, the text didn't mention anything about these noise terms.

2. Based on the explanation of Figure 2, the author's understanding of causal theory is fundamentally flawed. In section 3.2, the author suggests that the purpose of an intervention is to discover the true X->R relationship. Yet from the author's interpretation of Figure 2, do(X) only blocks the backdoor pathways in Figure 2, unable to eliminate front-door pathways, thus failing to establish the actual X->R relationship.

   To ascertain the veritable causal relationship between X->R, we must consider the front-door pathway X->K->R. Here, X is observable, hence we can compute P(K|do(X)) = P(K|X) (since there are no backdoor paths at this point). Furthermore, according to the section 2, K is unknown, so when calculating the causal effect of K->R, the backdoor pathway can be blocked by conditioning on X. We can then use the adjustment formula to calculate P(R|do(K)) by summing up over $\sum$ P(R|K, X)P(X).

3. It's unclear how the author's intervention is implemented specifically. For the intervention P(R|do(X)), it is necessary to observe changes in conditional probabilities under different X samplings to draw out the causal relationship between R and X. Do(X) is a shorthand, but the implementation needs to consider different situations like do(X=x1), do(X=x2), etc. However, it seems the author does not quite comprehend the genuine purpose of intervention.

In conclusion, I find it challenging to endorse the author's many explanations of causal relationships. The article tries to elucidate the motivations of its method from a causal modeling standpoint. However, many aspects are mere conjecture by the author, and there are many self-contradictions.

**Reproducibility:**

4: Could mostly reproduce the results, but there may be some variation because of sample variance or minor variations in their interpretation of the protocol or method.

**Reviewer Confidence:**

4: Quite sure. I tried to check the important points carefully. It's unlikely, though conceivable, that I missed something that should affect my ratings.

**Typos Grammar Style And Presentation Improvements:**

The paper seems to have a poor understanding of many definitions in causality theory. The author should try to gain a deeper understanding of SCMs and the concept of interventions.

---

> ### Author Rebuttal · Authors · 2023-08-28
>
> Thanks for your thoughtful comments! Please find the responses to the specific comments.
>
> > **Comment 1:** Figure 2 is not an SCM structure but a simple DAG (Directed Acyclic Graph) format.
>
> Thank you for your comments. The reason that Figure 2 does not include independent noise terms is that we assume that the context prior $C$ consists of unobserved prior and observed prior. Among them, we classify the independent noise terms into the unobserved prior.
>
> Based on your comments, we argue that such a setup may confuse the readers. In the revised version of the paper, we will add $N$->$X$, where $N$ denotes the independent noise terms.
>
>
>
> > **Comment 2:** do(X) only blocks the backdoor pathways in Figure 2, unable to eliminate front-door pathways, thus failing to establish the actual X->R relationship.
>
> Thank you very much for pointing out the problem. If we need to ascertain the veritable causal relationship between $X$->$R$, we do need to consider the front-door pathway $X$->$K$->$R$. Thank you for your detailed derivation.
>
> However, we argue that it is our statement "To pursue the real causality from $X$ to $R$" (Line 266) in section 3.2 that misleads you.
>
> Firstly, we argue that $R$ is determined by $X$ via two paths:
>
> 1) the direct $X$->$R$.
>
> 2) the mediation $X$->$K$->$R$.
>
> Then, in Line 266, we expect the expression "To pursue the real causality from X to R, including the two paths".
>
>
>
> > **Comment 3:** How the author's intervention is implemented?
>
> We are sorry that our description of the intervention method confused you. Below we first show the derivation of intervention (Eq. 5):
>
> $P(R \mid d o(x))  =\sum_{i=1}^{|C|} P(R \mid d o(X=x), C=c_{i}) P(C=c_{i} \mid d o(X=x)) \\  $
>
> $=\sum_{i=1}^{|C|} P(R \mid d o(X=x), C=c_{i}) P(C=c_{i})$
>
> $=\sum_{i=1}^{|C|} P(R \mid X=x, C=c_{i}) P(C=c_{i}) \\ $
>
> $=\sum_{i=1}^{|C|} \sum_{k} P(R \mid X=x, C=c_{i}, K=k) P(K=k \mid X=x, C=c_{i}) P(C=c_{i}) \\ $
>
> $=\sum_{i=1}^{|C|} P(R \mid X=x, C=c_{i}, K=g_{s}(x, c_{i})) P(C=c_{i}) \\ $
>
> $=\sum_{i=1}^{|C|}P\left(R | X, K=g_{s}\left(X, c_{i}\right)\right) P\left(c_{i}\right)$.
>
> Since the confounder $C$ is commonly hard to observe, we consider the entire label set as the partially observed children of the unobserved confounder.  We approximate it by designing a dictionary $D_{c} = \{c_{1}, c_{2},  \ldots, c_{|N|}\}$ as an $N \times d$ matrix, where $N$ represents the number of labels. Finally, the intervention is implemented as:
>
> $  P(R | d o(x))=\sum_{i=1}^{|C|}P\left(R | X, K=g_{s}\left(X, c_{i}\right)\right) P\left(c_{i}\right)$
>
> $ \approx \sum_{i=1}^{|N|}P\left(R | X, K=g_{s}\left(X, c_{i}\right)\right) P\left(c_{i}\right)$
>
> $=\sum_{i=1}^{|N|}\left[P\left(R | f_{s}(x_{j}, k_{i})\right) P\left(c_{i}\right)\right] $
>
> $= \sum_{i=1}^{|N|} \left[\textrm{softmax}(f_{s}(x_{j}, k_{i}))P\left(c_{i}\right)\right]$
>
> $\approx \textrm{softmax}( \sum_{i=1}^{|N|} f_{s}(x_{j}, k_{i})P\left(c_{i}\right) ).$
>
>
>
> > **Comment 4:** Statistical results relevant to the shortcut mentioned in Figure 1.
>
> We first provide the number of Intentional homicide cases and Manslaughter cases in the CAIL dataset we used:
>
> |              | Intentional homicide | Manslaughter |
> | ------------ | -------------------- | ------------ |
> | training set | 1877                 | 220          |
> | test set     | 442                  | 276          |
>
> We can find that in the training set, Intentional homicide occurs more frequently than Manslaughter, while there is not much difference between the two types of cases in the test set.
>
> In addition, we show the statistics on whether RNP and Inter-RAT correctly classify Intentional homicide cases and Manslaughter cases on the test set.
>
> |           | Intentional homicide (correct/wrong) | Manslaughter (correct/wrong) |
> | --------- | ------------------------------------ | ---------------------------- |
> | RNP       | 413/29                               | 165/111                      |
> | Inter-RAT | 421/21                               | 243/33                       |
>
> From the statistics, it can be seen that both RNP and Inter-RAT can achieve good results when predicting Intentional homicide, while Inter-RAT performs better than RNP when predicting Manslaughter. This observation demonstrates our causal intervention method can select more sufficient rationales for yielding results.
>
>
>
> > **Comment 5:** Why not elaborate the differences with these works in the related work section?
>
> We thank the reviewer for providing new related work. We refer to "Discovering invariant rationales for graph neural networks" [1] as DIR for short, and "Debiasing graph neural networks via learning disentangled causal substructure" [2] as DisC.
>
> Both DIR and DisC study the problem of rationalization in graph neural networks. They separate the graph as the rationale subgraphs and the non-rationale ones. Afterwards, interventions are implemented through counterfactual data augmentation methods to extract the faithful rationales.
>
> The major difference between our Inter-RAT and these methods is that we further focus on the limitations on the predefined sparsity in rationalization, where the sparsity constraint may result in spurious correlations between the extracted rationale and the predicted result.
>
> **References**
>
> [1] Ying-Xin Wu, Xiang Wang, An Zhang, Xiangnan He, and Tat seng Chua. Discovering invariant rationales for graph neural networks. In ICLR2022.
>
> [2] Fan, Shaohua, et al. Debiasing graph neural networks via learning disentangled causal substructure. In NeurIPS2022.

---

### Official Review · Reviewer_fKwz · 2023-08-11

**Typos Grammar Style And Presentation Improvements:** Not as far as I noticed.
**Soundness:** 5

**Excitement:**

4: Strong: This paper deepens the understanding of some phenomenon or lowers the barriers to an existing research direction.

**Missing References:**

The paper is well written; however, the review papers in the related works seems to mostly older as compared and rarely includes recent works such as below which have used causality to solve spurious relationships.

Niu, Yulei, et al. "Counterfactual vqa: A cause-effect look at language bias." Proceedings of the IEEE/CVF Conference on Computer Vision and Pattern Recognition. 2021.
Vosoughi, Ali, et al. "Unveiling Cross Modality Bias in Visual Question Answering: A Causal View with Possible Worlds VQA." arXiv preprint arXiv:2305.19664 (2023).

**Paper Topic And Main Contributions:**

The paper talks about rationalization models that have been prior proposed by INVRAT as  means of inferring based on relating outcomes to some rationals, being said as X -> R -> Y. Here Y is outcome and R is rational which is based on input and it is modeled by P(R|X).

What authors put in here is to propose two backdoor additions to INVRAT to propose a method that identified causal paths by removing confounder effects, call it Inter-RAT and show that the method does relatively better than existing literature in this sense.

**Questions For The Authors:**

How authors would expand it to more recent large models in vision and language?

**Reasons To Accept:**

The paper is about a method that seems to be generalizable and scalable, as it uses causality and reasoning to improve their tasks. Therefore, there is hope for wider future applications in the dawn of large foundational models. The paper may be interesting to EMNLP audience.

**Reasons To Reject:**

The study sometimes talks about benchmarks that have been itemized by authors, so how authors assure that these samples are fairly selected and no cherry picking?

Example in line 603
---Second, as CAIL does not provide annotated...

**Reproducibility:**

4: Could mostly reproduce the results, but there may be some variation because of sample variance or minor variations in their interpretation of the protocol or method.

**Reviewer Confidence:**

4: Quite sure. I tried to check the important points carefully. It's unlikely, though conceivable, that I missed something that should affect my ratings.

---

> ### Author Rebuttal · Authors · 2023-08-28
>
> We appreciate your comments! We are very glad that you have a positive impression on our work. To address your concerns, below we provide the responses as follows:
>
> > **Comment 1:** The study sometimes talks about benchmarks that have been itemized by authors, so how authors assure that these samples are fairly selected and no cherry picking?
>
> We are sorry that our lack of detail on selecting and evaluating the human evaluation dataset confuses you. Below we describe how the dataset is selected and evaluated.
>
> To ensure fairness, we randomly sample the samples in the test set. We do not perform any post-processing on the dataset obtained from this sampling.
>
> For the human evaluation, we shuffle all the results generated by the RNP, INVRAT, and Inter-RAT, and then distribut them to legal professionals for annotation.
>
> The annotators do not know which method generated the samples they evaluate. Of course, for the purpose of subsequent aggregation of results, we (the authors) know which sample to be evaluated is generated by which method, but we do not participate in the evaluation.
>
> Finally, we promise to release our human evaluation dataset.
>
>
>
> > **Comment 2:** How authors would expand it to more recent large models in vision and language?
>
> Our approach can be used to provide explanations of text classification tasks. For large language models (LLMs), in the autoregressive generation, we can treat the prediction of each token as a text classification task, where the number of categories is equal to the size of the vocab. By generating a corresponding explanation for each token, we can gradually explain the output of a LLM.
>
> However, applying Inter-RAT to the explanation of LLMs is still a challenging task. We will continue to research and explore it as the focus of our future work.
>
>
>
> > **Comment 3:** Missing References
>
> Thank you very much for the related work, which we will cite in the revised version of the paper.

---

### Official Review · Reviewer_YC9Z · 2023-08-11

**Soundness:** 4

**Excitement:**

4: Strong: This paper deepens the understanding of some phenomenon or lowers the barriers to an existing research direction.

**Paper Topic And Main Contributions:**

The paper focuses on the selective rationalization task in DNNs, aiming to identify significant features impacting predictions. The paper introduces an interventional rationalization (Inter-RAT) method, applying causal interventions to remove spurious correlations. Experimental results on real-world datasets validate the effectiveness of the approach. As a result, the proposed method achieves SOTA on many datasets.

**Questions For The Authors:**

There are some reasons to reject, but I am considering changing the scores depending on the response.

A. I may have missed a detail, but is there a reason why HardKuma was not subjected to human evaluation in Table 4? I would like to know the reason for excluding the model that performs well compared to other baselines.

B. Looking at the baselines, there is no method published from 2022 to the present. Is there really no paper that has been published? Is there a reason why the model that appeared in [1] was not included as a baseline? Since 2022, LLMs have rapidly advanced, and models like ChatGPT seem to be able to simultaneously act as both a selector and a predictor even in zero-shot and few-shot settings. Have you attempted to add LLMs to the baseline?

[1] Li et al. Unifying Model Explainability and Robustness for Joint Text Classification and Rationale Extraction, AAAI 2021.

**Reasons To Accept:**

1) It recorded the SOTA across various datasets.
2) The paper is well-written, making it easy to read, including problem definition, method proposal, and result analysis.
3) The experimental results show a trend that is consistent with human evaluation.

**Reasons To Reject:**

1) I wish there was a more detailed explanation for Figure 2. The text is small and lacks content, making it hard to read.
2) HardKuma baseline is missing in the Human evaluation (Table 4).
3) The baseline is inadequate. There is no method published from 2022 to the present

**Reproducibility:**

4: Could mostly reproduce the results, but there may be some variation because of sample variance or minor variations in their interpretation of the protocol or method.

**Reviewer Confidence:**

4: Quite sure. I tried to check the important points carefully. It's unlikely, though conceivable, that I missed something that should affect my ratings.

---

> ### Author Rebuttal · Authors · 2023-08-28
>
> Thanks for your important comments! Below we conduct additional experiments and provide the responses:
>
> >**Comment 1:** I wish there was a more detailed explanation for Figure 2. The text is small and lacks content, making it hard to read.
>
> Thank you for your comments, and we will enlarge the font and add content to Figure 2 in the revised version of the paper. Specifically, we will provide the following explanation in the caption of Figure 2:
>
> (a) SCM for rationalization. (b) SCM for the selector, which offers a ﬁne-grained causal relationship between $R$ and $C$ with a mediator $K$. (c) Intervention on the selector, where $C$ -> $X$ is cut off  and the confounder $C$ is stratified into pieces $C = \{c\_{1}, c\_{2},  \dots, c\_{|C|}\}$. (d) SCM for the predictor, which describes a ﬁne-grained causal relationship between $Y$ and $C$ with a mediator $H$.
>
> >**Comment 2:** The baseline is inadequate. There is no method published from 2022 to the present.
>
> Thank you for bringing baselines in [1], including IB, FRESH, IB (semi-supervised, 25%) , Weakly- & Semi-supervised, AT-BMC and Pipeline.
>
> Current rationalization methods can be classified into three groups, i.e. unsupervised rationalization, semi-supervised rationalization and supervised rationalization based on whether the training set contains human annotated rationale or not. In this paper, our method belongs to the category of unsupervised rationalization.
>
> For baselines in [1], the classification is shown in the following Table:
>
> | unsupervised rationalization    | IB，FRESH                                             |
> | ------------------------------- | ----------------------------------------------------- |
> | semi-supervised rationalization | IB (semi-supervised, 25%) ，Weakly- & Semi-supervised |
> | supervised rationalization      | AT-BMC，  Pipeline                                    |
>
> Therefore, we choose IB, FRESH as our new baselines. Besides, we add DARE [2] which is published in 2022 as our new baseline.
>
> From Table, we can find Inter-RAT outperforms the baselines consistently in ﬁnding correct rationales.
>
> |           |      |              |  Appearance  |              |              |    Aroma     |              |              |    Palate    |              |
> | :-------: | :--: | :----------: | :----------: | :----------: | :----------: | :----------: | :----------: | :----------: | :----------: | :----------: |
> |  method   |  a   |   Presion    |    Recall    |      F1      |   Presion    |    Recall    |      F1      |   Presion    |    Recall    |      F1      |
> |    RNP    | 0.1  |   32.4±0.5   |   18.6±0.3   |   23.6±0.4   |   44.8±0.4   |   32.4±0.7   |   37.6±0.5   |   24.6±0.5   |   23.5±0.5   |   24.0±0.5   |
> | HardKuma  | 0.1  |   53.6±0.1   |   28.7±0.1   |   37.4±0.1   |   29.3±1.4   |   25.9±3.8   |   27.3±2.1   |   7.7±0.1    |   6.0±0.1    |   6.8±0.1    |
> |  INVRAT   | 0.1  |   42.6±0.7   |   31.5±0.6   |   36.2±0.6   |   41.2±0.3   |   39.1±2.8   |   40.1±1.6   | **34.9±1.5** |   45.6±0.2   |   39.5±1.0   |
> |    IB     | 0.1  |   50.5±0.2   |   29.7±0.5   |   37.4±0.4   |   43.5±0.3   |   39.2±0.5   |   41.3±0.1   |   29.9±0.2   |   34.2±0.7   |   31.9±0.2   |
> |   FRESH   | 0.1  |   51.4±0.3   |   29.5±0.5   |   37.5±0.4   |   42.8±0.4   |   40.4±0.3   |   41.5±0.3   |   29.3±0.5   |   32.7±0.2   |   30.9±0.2   |
> |   DARE    | 0.1  |   63.9±0.1   |   42.8±0.2   |   51.3±0.1   |   50.5±0.1   |   44.8±0.2   |   47.5±0.2   |   33.1±0.4   |   45.8±0.1   |   38.4±0.2   |
> | Inter-RAT | 0.1  | **66.0±0.4** | **46.5±0.8** | **54.6±0.7** | **55.4±0.9** | **47.5±0.6** | **51.1±0.8** |   34.6±0.8   | **48.2±0.4** | **40.2±0.5** |
> |           |      |              |              |              |              |              |              |              |              |              |
> |    RNP    | 0.2  |   39.4±0.4   |   44.9±0.1   |   42.0±0.2   |   37.5±0.1   |   51.9±0.7   |   43.5±0.3   |   21.6±0.4   |   38.9±0.5   |   27.8±0.4   |
> | HardKuma  | 0.2  | **64.9±0.9** |   69.2±1.0   |   67.0±0.8   |   37.0±1.3   |   55.8±1.9   |   44.5±1.5   |   14.6±0.3   |   22.3±0.8   |   17.7±0.4   |
> |  INVRAT   | 0.2  |   58.9±0.4   |   67.2±2.3   |   62.8±1.1   |   29.3±1.0   |   52.1±0.6   |   37.5±0.6   |   24.0±1.3   |   55.2±2.3   |   33.5±1.6   |
> |    IB     | 0.2  |   59.3±0.4   |   69.0±0.2   |   63.8±0.2   |   38.6±0.1   |   55.5±0.7   |   45.6±0.1   |   21.6±0.2   |   48.5±0.4   |   29.9±0.2   |
> |   FRESH   | 0.2  |   59.7±0.3   |   70.1±0.2   |   64.5±0.2   |   39.4±0.3   |   55.4±0.2   |   46.0±0.1   |   23.7±0.2   |   50.5±0.1   |   32.3±0.2   |
> |   DARE    | 0.2  |   63.7±0.2   |   71.8±0.8   |   67.5±0.5   |   41.0±0.2   |   61.5±0.2   |   49.3±0.3   |   24.4±0.1   |   54.9±0.8   |   33.8±0.1   |
> | Inter-RAT | 0.2  |   62.0±0.5   | **76.7±1.7** | **68.6±0.4** | **44.2±0.1** | **65.4±0.2** | **52.8±0.1** | **26.3±0.6** | **59.1±0.8** | **36.4±0.7** |
> |           |      |              |              |              |              |              |              |              |              |              |
> |    RNP    | 0.3  |   24.2±0.4   |   41.2±0.8   |   30.5±0.5   |   27.1±0.3   |   55.7±0.8   |   36.4±0.4   |   15.4±0.4   |   42.2±0.9   |   22.6±0.5   |
> | HardKuma  | 0.3  |   42.1±0.3   |   82.4±1.4   |   55.7±0.5   |   24.6±0.1   |   57.7±0.6   |   34.5±0.2   |   21.7±0.1   |   49.7±0.4   |   30.2±0.1   |
> |  INVRAT   | 0.3  |   41.5±0.4   |   74.8±0.3   |   53.4±0.3   |   22.8±1.6   |   65.1±1.7   |   33.8±1.8   |   20.9±1.1   | **71.6±0.4** |   32.3±1.3   |
> |    IB     | 0.3  |   40.2±0.1   |   81.5±0.2   |   53.9±0.2   |   27.9±0.2   |   59.2±0.3   |   37.9±0.1   |   19.1±0.7   |   59.0±0.9   |   28.9±0.6   |
> |   FRESH   | 0.3  |   41.4±0.3   |   81.2±0.2   |   54.8±0.3   |   29.5±0.4   |   60.9±0.2   |   39.7±0.3   |   21.1±0.2   |   60.2±0.3   |   31.2±0.2   |
> |   DARE    | 0.3  |   45.5±0.2   |   80.6±0.2   |   58.1±0.1   |   32.7±0.2   |   68.2±0.3   |   44.2±0.1   |   19.7±0.6   |   70.5±0.4   |   30.8±0.7   |
> | Inter-RAT | 0.3  | **48.1±0.7** | **82.7±0.5** | **60.8±0.4** | **37.9±0.7** | **72.0±0.1** | **49.6±0.7** | **21.8±0.1** |   66.1±0.8   | **32.8±0.1** |
>
> >**Comment 3:** HardKuma baseline is missing in the Human evaluation & there a reason why HardKuma was not subjected to human evaluation in Table 4? & Have you attempted to add LLMs to the baseline?
>
> Thank you for your suggestion. Since INVRAT is the baseline for our main comparison, we do not conduct a human evaluation of HardKuma.
>
> To clarify your confusion, we add the human evaluation of HardKuma in Table below. In addition, due to time constraints, we add the human evaluation of DARE, where DARE achieves competitive results in extracting rationales.
>
> It is worth noting that the similar human evaluation is provided in DARE, but its $\alpha$ is set differently from ours, where DARE sets $\alpha$ to 0.14 and our Inter-RAT is set to 0.2. For consistency, we replicate DARE with  $\alpha$ = 0.2.
>
> From Table, we can see that both HardKuma and DARE do not achieve as good results as Inter-RAT, which also shows the effectiveness of our method.
>
> Also, based on your suggestion, we additionally add chatgpt for human evaluation.
>
> We can observe that chatgpt achieves competitive results on both Completeness and Fluency. But it does not perform well on Usefulness. Because even though we ask chatgpt to extract short tokens and sentences as rationale, it still tends to extract longer sentences, which is beneficial for the Fluency metric. However, for the Usefulness metric, some redundant tokens will be extracted as well. Therefore chatgpt does not perform well on the Usefulness.
>
> | Methods   | Usefulness (U) | Completeness (C) | Fluency (F) |
> | --------- | -------------- | ---------------- | ----------- |
> | RNP       | 3.85           | 3.28             | 3.34        |
> | INVRAT    | 3.88           | 3.42             | 3.41        |
> | HardKuma  | 3.78           | 3.33             | 3.39        |
> | DARE      | 4.57           | 3.89             | 4.19        |
> | chatgpt   | 4.01           | 4.08             | 4.34        |
> | Inter-RAT | 4.52           | 4.10             | 4.25        |
>
>
>
> This is our prompt for chatgpt:
>
> Suppose you are a data tagging assistant and I will provide you with some case facts. Please follow the following prompts to make your predictions:
>
> 1.Please predict the charge of the given case facts.
>
> 2.Please extract one or more consecutive sentences and tokens from the case facts to support your prediction and keep the extracted sentences and tokens as short as possible. Please note that you can only extract from the given case facts.
>
> 3.Only output a python dictionary format to me, e.g. {'label':intentional homicide, 'rationale':extract sentence}.
>
>
>
> **References**
>
> [1] Dongfang Li, Baotian Hu, Qingcai Chen, Tujie Xu, Jingcong Tao, Yunan Zhang. Unifying Model Explainability and Robustness for Joint Text Classication and Rationale Extraction. In AAAI2021.
>
> [2] Linan Yue, Qi Liu, Yichao Du, Yanqing An, Li Wang, Enhong Chen. DARE: Disentanglement-Augmented Rationale Extraction.  In NeurIPS2022.

---

### Meta-Review · Area_Chair_WAPa · 2023-09-18

**Recommendation:** 4

**Metareview:**

This paper proposes Inter-RAT, a method based on causal analysis to enhance the explainability of the neural networks. By eliminating spurious correlations using the causal intervention, the proposed method outperformed previous baselines on multiple datasets. Reviewers all agreed on the potential utility of the proposed method in the NLP community. However, a strong concern remained regarding the improper understanding of structural causal model (SCM), which the paper relied on developing the proposed method, and hence the potentially unreliable validity of the discovered causal relationship.

---

### Decision · Program_Chairs · 2023-10-07

**Decision:**

Accept-Main

**Comment:**

This paper proposes Inter-RAT, a method based on causal analysis to enhance the explainability of the neural networks. By eliminating spurious correlations using the causal intervention, the proposed method outperformed previous baselines on multiple datasets. Reviewers all agreed on the potential utility of the proposed method in the NLP community. However, a strong concern remained regarding the improper understanding of structural causal model (SCM), which the paper relied on developing the proposed method, and hence the potentially unreliable validity of the discovered causal relationship.